# How Environmental Regulation Affects Industrial Green Total Factor Productivity in China: The Role of Internal and External Channels

**Guihuan Yan [1,2], Liming Jiang [3,*] and Chongqing Xu [1,4]**

1    Faculty of Environmental Science and Engineering, Qilu University of Technology (Shandong Academy of Sciences), Jinan 250000, China
2    Ecology Institute of Shandong Academy of Sciences, Jinan 250000, China
3    Faculty of Economics and Management, Qilu University of Technology (Shandong Academy of Sciences), Jinan 250000, China
4    Shandong Technology Innovation Center of Carbon Neutrality, Jinan 250000, China
*    Correspondence: limingjiang158@163.com; Tel.: +86-158-65171661

**Abstract:** Many nations have enacted diverse environmental control regulations to address environmental and climate concerns. Analyzing how environmental regulation affects industrial green total factor productivity can aid in creating appropriate environmental regulation laws and realizing peaceful coexistence between man and nature. Based on the panel data of various provinces in China from 2011 to 2019, this paper used the data envelopment analysis method to measure the industrial green total factor productivity and then used the system generalized method of moments model to empirically study the differential effect of heterogeneous environmental regulation on China's industrial green total factor productivity. In addition, this paper also conducted a test of internal and external mechanisms. The statistics show that environmental regulation can boost the growth of industrial green total factor productivity and pass the robustness test. Innovation ability is the external mechanism variable of environmental regulation acting on industrial green total factor productivity. Regulation can improve industrial productivity and significantly suppress industrial pollution emissions, but market-based environmental regulations do not have an effective impact on carbon emissions. Environmental regulations in economically developed regions can promote the growth of industrial green total factor productivity, but for financially backward areas, market-based environmental regulation inhibits the promotion of industrial green total factor productivity, while command-and-control environmental regulation is not helpful for industrial green total factor productivity.

**Keywords:** green total factor productivity; environmental regulation; system generalized method of moments; transmission mechanism; carbon emission

## 1. Introduction

China's proportion of the world's gross domestic product (GDP) has climbed from 1.74% in 1978 to 18.5% in 2021, representing a period of significant economic development. However, China's industrialization has always been accompanied by high input consumption and pollution emissions. According to the calculation of energy data in 2019, China's industry generated just 32% of the added value, with 68% of carbon dioxide emissions and 65% of energy consumption. China's former growth model has mainly relied on a significant input of production elements. At the same time, it needs to emit many pollutants and greenhouse gases into the environment. It can be seen that the level of green development is relatively low [1]. The extensive development model is becoming increasingly unsustainable as environmental quality deteriorates and marginal returns on production variables decline. Therefore, it is necessary to guide the industry toward greening and big value-added improvement.

Many countries have implemented environmental regulation policies to solve environmental problems [2]. For example, the United States has established organic laws of environmental protection agencies and the pollution rights trading market, France has boosted investment in environmental protection, and the Netherlands has established an environmental protection tax. China has increased its focus on environmental conservation and established laws to prevent environmental degradation in recent years. For example, China has established policies such as dual energy policies, environmental protection taxes, and carbon trading markets [3,4]. However, additional research is required to determine whether these environmental control policies can increase industrial green total factor productivity and how they affect industrial green total factor productivity. Based on this, this paper analyzed the impact of environmental regulation on industrial green total factor productivity and tested the internal and external transmission mechanisms. The framework of this paper is designed as follows. The second part is a literature review on environmental regulation and industrial green total factor productivity. The third part is the data source and methods about the calculation method of industrial green total factor productivity, and the main econometric regression model. The fourth part is the empirical analysis results. The fifth part is the discussion. The sixth part is the conclusions and policy recommendations.

## 2. Literature Review

Industrial green total factor productivity, often known as industrial GTFP, is the input–output efficiency of the industrial sector that includes undesirable outcomes [5]. It centrally symbolizes the degree of advanced industrial greening. Improving industrial GTFP is crucial to achieving sustainable development in China. There are many methods for calculating industrial GTFP, data envelopment analysis being one of them. The data envelopment analysis approach, also known as the DEA method, evaluates efficiency based on a number of input–output indicators using linear programming. Because the DEA approach can take into account all undesirable output, it has been widely adopted. The DEA method was first given by Charnes et al. [6], and then, Chung et al. proposed the Directional Distance Function (DDF), which made pollution an undesired output to be considered in the production process. As a result, it has become possible to avoid the disadvantages of taking environmental pollution as an input factor in the past, and it can more truly reflect the production process. To solve the slack variable problem, Tone proposed the Slack-Based Measure (SBM) model [7]. To solve the problem of inter-temporal incomparability of productivity indices, Pastor et al. proposed the concept of Global Malmquist (GML) [8], which uses the sum of each period as a reference set to calculate productivity. Until now, the DEA has developed into a reasonably sophisticated GTFP computation methodology. Numerous academics have performed multiple calculations on GTFP using the DEA model. For example, Vivek et al. calculated the GTFP of the Swedish paper industry [9], and Zhang et al. calculated the GTFP of each province in China [10].

As a typical public good, the environment will cause manufacturers to overuse it, cause negative externalities of production, and ultimately lead to social inefficiency and environmental pollution problems [11,12]. Based on this theory, Pigou proposed a scheme of levying taxes to realize the internalization of external problems and solve the problem of environmental pollution. Some scholars also put forward the view of "establishing a market" [13] so that emissions or pollutant discharge permits can be publicly traded in the market to achieve green economic development. Some scholars believe that the government should use laws and regulations to control environmental pollution [14]. These methods of controlling environmental pollution can be collectively referred to as environmental regulation, which are some tools used by the government to reduce environmental pollution [15,16].

When studying how environmental management affects industrial GTFP, the place of innovation capacity in the model should be clarified, which is the test of the "Porter Hypothesis" [17]. It will inevitably involve the "innovation compensation effect" and

the "following cost effect", which requires innovation capacity to be put into the model, which is what we often call the transmission mechanism test. To separate it from the internal conduction mechanism later in the paper, we call this conduction mechanism the external conduction mechanism. A number of scholars have studied the effect of environmental control on innovation. For example, when Umar et al. analyzed some non-financial enterprises in Asia, they found that regulation can promote productivity progress and technological innovation [18]. From the perspective of the environmental protection target responsibility system, Hu et al. found that environmental regulation only benefit innovation in China's developed regions [19]. So many scholars have directly described the relationship between them, and few studies have incorporated innovation capability into the framework of environmental management affecting industrial GTFP.

Most scholars took industrial GTFP as a whole when it was the subject of study [20,21]. However, the industrial GTFP describes the degree of industrial intensification and green growth in the region, which reflects not only the changes in industrial output value but also the pollutants and carbon emissions, how to smartly design environmental regulations so the economy can grow steadily while reducing emissions of pollutants and carbon dioxide. This requires a more in-depth study of the interaction of regulation and industrial GTFP. Despite the fact that several studies have decomposed industrial GTFP into scale efficiency and technical efficiency [22], few scholars have examined the internal transmission mechanism from the perspectives of the industrial economy and environment. Clarifying this relationship can help us better design policies. This paper calls it an internal transmission mechanism because environmental regulation does not affect industrial GTFP through external forces such as innovation but changes the internal state of industrial GTFP, which in turn changes industrial GTFP.

To sum up, on the one hand, although previous studies have systematically described how environmental regulation affects innovation, they have not analyzed the external mechanism of environmental regulation affecting industrial GTFP. How exactly does environmental regulation affect industrial GTFP? What is the place of innovation capability in the model? On the other hand, previous studies have analyzed the internal conduction mechanism by which environmental regulation affects industrial GTFP. Incorporating technical innovation capability into the model and assessing how environmental regulation affects industrial GTFP constitutes one of the paper's innovations. The second innovation of this paper is that the industrial GTFP has been decomposed into three parts: industrial output value, pollution emission, and carbon dioxide emission, in order to study the effect of heterogeneous environmental regulation on each decomposed part and then analyze the environmental regulation's impact on the internal conduction mechanisms of industrial GTFP.

## 3. Methodology

### 3.1. Industrial GTFP

3.1.1. Measurement of Industrial GTFP

Since the SBM direction distance function can not only reflect the undesired output but also measure the slack part of the production when measuring the production efficiency [23,24], this paper has used the SBM function combined with the GML index to measure. The GML index represents its growth, not an absolute value [25,26]. To better match the research purpose, this article has taken 2010 as the base year and multiplied the industry GML index to obtain the industry GTFP of each year.

$$P^G(x) = \left\{ (y^t, b^t) : \sum_{t=1}^{T} \sum_{k=1}^{K} z_k^t y_{km}^t \geqslant y_{km}^t, \forall m; \sum_{t=1}^{T} \sum_{k=1}^{K} z_k^t b_{ki}^t = b_{ki}^t, \forall i; \\ \sum_{t=1}^{T} \sum_{k=1}^{K} z_k^t x_{kn}^t \leqslant x_{kn}^t, \forall n; \sum_{k=1}^{K} z_k^t = 1, z_k^t > 0, \forall k \right\} \tag{1}$$

The above formula reflects the global production possibility set $P^G = P^1 \cup P^2 \cup \ldots \cup P^T$. Assume that the production process requires input unit $X = (x_1, x_2, \ldots, x_N) \in R_N^+$, desirable output units $Y = (y_1, y_2, \ldots, y_M) \in R_M^+$, and undesirable output unit $B = (b_1, b_2, \ldots, b_M) \in R_B^+$. Then, the SBM function form is as follows:

$$S_V^G\left(x^{t,k'}, y^{t,k'}, b^{t,k'}, g^x, g^y, g^b\right) = \max_{s^x, s^y, s^b} \frac{1}{2N} \sum_{n=1}^{N} \frac{s_n^x}{g_n^x} + \frac{\left(\sum_{m=1}^{M} \frac{s_m^y}{g_m^y} + \sum_{i=1}^{I} \frac{s_i^b}{g_i^b}\right)}{2(M+I)}$$

$$\text{s.t.} \sum_{t=1}^{T} \sum_{k=1}^{K} z_k^t x_{kn}^t + s_n^x = x_{k'n}^t, \forall n; \sum_{t=1}^{T} \sum_{k=1}^{K} z_k^t y_{km}^t - s_m^y = y_{k'm}^t, \forall m; \sum_{t=1}^{T} \sum_{k=1}^{K} z_k^t b_{ki}^t + s_i^b = b_{k'i}^t, \forall i \qquad (2)$$

$$\sum_{k=1}^{K} z_k^t = 1, z_k^t \geqslant 0, \forall k; s_n^x \geqslant 0, \forall n; s_m^y \geqslant 0, \forall m; s_i^b \geqslant 0, \forall i$$

The above formula is the SBM directional distance function constructed in this paper, where $g$ represents the direction vector of each element, and $s$ represents the relaxation vector. Based on the research of other scholars [27], this paper constructs the following GML index.

$$GML_t^{t+1} = \frac{1 + S_V^G(x^t, y^t, b^t; g)}{1 + S_V^G(x^{t+1}, y^{t+1}, b^{t+1}; g)} \qquad (3)$$

3.1.2. Indicators Selection for Industrial GTFP

On the basis of learning from previous research, the input indicators of this paper have included industrial energy consumption, net industrial fixed assets and labor force [28]. Industrial value added is not suitable as an expected output variable as inputs contain intermediate products of energy, the industrial output value data in the statistical yearbook were only counted up to 2011, and the sales output value was counted up to 2017. Therefore, this paper has used the output value of industries above the designated size (operating income of industries above the designated size in 2018 and 2019) as the expected output indicator.

Undesirable outputs include industrial pollution emissions and industrial carbon dioxide. Industrial pollution emissions consist of wastewater, waste gas, and solid waste. Industrial wastewater has been represented by chemical oxygen demand and ammonia nitrogen emissions using the entropy technique. Using the research of Qiu et al. as a reference [29], this paper uses sulfur dioxide emissions to represent industrial waste gas. Industrial solid waste has utilized statistical yearbook data directly. The weight of the three industrial wastes has been determined using the entropy method. This article has calculated industrial carbon dioxide emissions based on each province's energy balance sheets. Since carbon dioxide is only produced when energy is burned, the net industrial energy consumption was employed for this study. Formula (4) is the specific technique of calculation.

$$CO_2 = \sum_{i=1} CO_2 = \sum_{i=1} E_i \times NCV_i \times COF_i \times CEF_i \times \frac{44}{12} \qquad (4)$$

Among them, *NCV* is the average low calorific value of fossil fuels, *COF* is the carbon oxidation rate, *CEF* is the carbon content per unit calorific value, and *E* represents the consumption of various fossil fuels. The statistics have come from the data of Chinese provinces from 2010 to 2019 (since the data of some provinces are not available, the research scope of this paper only includes 30 provinces in China). The relevant data are obtained from the China Industrial Statistical Yearbook, China Energy Statistical Yearbook, IPCC National Greenhouse Gas Emission Inventory Guidelines and provincial statistical yearbooks, and some missing data have been processed by linear interpolation.

*3.2. Regression Models and Variable Selection*

Since the productivity measured by the GML method has a serial correlation [30], to better explore the impact of GTFP in previous years, this paper has introduced the

system generalized method of moments model, which is also known as the GMM model. Compared with the differential GMM, the system GMM can better reduce the endogenous impact. Therefore, this paper has chosen the system GMM as the regression model of this paper. The specific regression model is as follows:

$$GTFP_{it} = \alpha + \chi GTFP_{it-1} + \beta er_{it} + \sum \lambda_i X_{it} + \mu_i + \gamma_t + \varepsilon_{it} \tag{5}$$

where GTFP represents industrial GTFP; $GTFP_{it-1}$ represents industrial GTFP with a lag of one order; *er* represents environmental regulation, which can be divided into command-and-control environmental regulation and market-based environmental regulation; $X$, $\mu$, $\gamma$ and $\varepsilon$ represent control variables, individual effects, time effects, and random disturbance terms, respectively.

In order to analyze how environmental regulation causes changes in industrial GTFP through innovation capability, this paper has introduced the innovation capability (*tech*) of industrial enterprises as an external mechanism variable into the model, as follows:

$$tech_{it} = \alpha + \chi tech_{it-1} + \beta er_{it} + \sum \lambda_i X_{it} + \mu_i + \gamma_t + \varepsilon_{it} \tag{6}$$

$$GTFP_{it} = \alpha + \chi GTFP_{it-1} + \beta er_{it} + \delta tech_{it} + \sum \lambda_i X_{it} + \mu_i + \gamma_t + \varepsilon_{it} \tag{7}$$

So as to verify and explore the internal principles of heterogeneous regulation for industrial GTFP, this paper has introduced industrial production efficiency (*effi*), industrial pollutant emission level (*poll*) and industrial carbon emission level (*carb*) as internal mechanism variables into the model. Different from the external conduction mechanism, the internal conduction mechanism variables are the components of industrial GTFP. Greater industrial productivity correlates to greater industrial GTFP, whereas increased pollution and carbon emissions correlate to decreased industrial GTFP, so there is no need to use the classical three-step method to analyze the internal conduction mechanism. It is enough to analyze whether environmental regulation is helpful to each internal transmission mechanism variable. For details, please refer to the following model:

$$\begin{aligned} effi_{it} &= \alpha + \chi effi_{it-1} + \beta er_{it} + \sum \lambda_i X_{it} + \mu_i + \gamma_t + \varepsilon_{it} \\ poll_{it} &= \alpha + \chi poll_{it-1} + \beta er_{it} + \sum \lambda_i X_{it} + \mu_i + \gamma_t + \varepsilon_{it} \\ carb_{it} &= \alpha + \chi carb_{it-1} + \beta er_{it} + \sum \lambda_i X_{it} + \mu_i + \gamma_t + \varepsilon_{it} \end{aligned} \tag{8}$$

This paper has taken the industrial GTFP of each province as the dependent variable of the model. The key independent variables of this paper have included command-and-control environmental regulation and market-based environmental regulation. Although China started a pilot carbon trading market in some regions in 2011, a nationwide carbon trading market was not established until 2021. There is only one year of data for nationwide carbon trading. Therefore, this paper does not study carbon-related environmental regulations, so the market-based environmental regulation in this paper has been only expressed by the logarithm of the ratio of the pollutant discharge fee (environmental protection tax in 2018 and later) to the total discharge of the three industrial wastes (*lnmar*). The higher the value, the higher the unit pollutant charge and the stricter the environmental control. In previous studies, the number of environmental laws was mostly used to represent command-and-control environmental regulations, but in recent years, the lack of environmental laws data has become more serious. In this paper, the logarithm of the ratio of environmental administrative penalty to pollutant (*lncom*) discharge is used to represent the command-and-control environmental regulation. Higher values indicate stricter environmental regulations.

This paper has used industrial technology innovation capability (*tech*) instead of R&D as the external mechanism variable, because R&D only reflects the investment in research and development and cannot directly reflect the real technical ability. Therefore, this paper calculated the technological innovation ability of each province using the SBM function, with the capital stock of industrial R&D, the equivalent of industrial R&D workers, and

the number of industrial R&D projects serving as input indicators. The output indicator is the total number of inventive patents with industrial validity. This paper has used the perpetual inventory method to estimate the R&D capital stock [31]. Industrial production efficiency (*effi*), industrial pollutant discharge (*poll*) and industrial carbon emission (*carb*) have been selected as internal transmission mechanism variables. Industrial production efficiency has been calculated using the SBM method, the input index was the same as that of GTFP, and the output index has only included the industrial sales output value. The discharge of industrial pollutants has been expressed by the ratio of three industrial wastes and the number of enterprises above the designated size, and the weight has been calculated by the entropy method. The ratio of industrial carbon dioxide emissions to the number of regulated businesses has been used to quantify industrial carbon emissions.

The following control variables have been selected to control the disturbance of other variables based on previous research: the degree of economic development (*lnpgdp*) has been measured by the logarithm of per capita GDP, foreign capital dependence (*fdi*) has been expressed as the proportion of foreign direct investment in GDP, road traffic (*infra*) has measured in kilometers of expressways owned per square kilometer, the level of capital deepening (*lncappeo*) has been expressed as the logarithm of the net fixed assets owned by a single labor force, the ownership structure (*own*) has been represented by the proportion of the operating income of industrial state-owned holding enterprises above the designated size, the industry concentration (*big*) has been expressed by the proportion of the operating income of large enterprises, and the R&D investment (*rd*) has been expressed as the flow of internal expenditures of industrial R&D funds above the designated size. All nominal variables have price effects removed to eliminate price effects. In addition to the yearbooks mentioned above, the data sources also include the Wind database and the National Bureau of Statistics database. Table 1 shows the details of each variable.

**Table 1.** The details of each variable.

| Variables | Max | Mean | Sd | Min |
|-----------|-----|------|-----|-----|
| GTFP | 1.494 | 1.079 | 0.107 | 0.840 |
| mar | 150.1 | 8.657 | 13.60 | 0.411 |
| com | 35.43 | 1.092 | 3.435 | 0.00565 |
| tech | 1 | 0.315 | 0.198 | 0.0436 |
| pgdp | 148,002 | 49,581 | 25,845 | 14,866 |
| fdi | 0.121 | 0.0213 | 0.0191 | 0.000096 |
| infra | 0.126 | 0.0311 | 0.026 | 0.000901 |
| cappeo | 922.8 | 51.58 | 60.97 | 11.86 |
| own | 0.836 | 0.357 | 0.176 | 0.0959 |
| big | 0.769 | 0.435 | 0.124 | 0.142 |
| rd | 0.0186 | 0.00809 | 0.00346 | 0.00276 |
| effi | 1 | 0.474 | 0.168 | 0.194 |
| carb | 17.09 | 4.072 | 3.591 | 0.378 |
| poll | 0.768 | 0.104 | 0.133 | 0.00143 |

## 4. Results

### 4.1. Benchmark Regression

The basic statistical results are represented in Table 2 by the outcomes of mixed regression, fixed effect regression, and systematic GMM regression. It shows that the AR (2) and the probability *p* values of Sargan are not statistically significant, indicating that the model does not have second-order autocorrelation, the instrumental variables are effective, and the system GMM model is suitable. It also shows that the first-order lag term of industrial GTFP promotes current GTFP, and it passes the 1% significance test, indicating that industrial GTFP possesses autocorrelation characteristics, and the effect is positive: the current value will promote the growth of the value for the next period.

**Table 2.** Benchmark regression.

| Variables | Mixed Regression (GTFP) | | Fixed Regression (GTFP) | | Dynamic Panel Regression (GTFP) | |
|---|---|---|---|---|---|---|
| | Model 1 | Model 2 | Model 3 | Model 4 | Model 5 | Model 6 |
| L.GTFP | | | | | 0.949 *** | 0.955 *** |
| | | | | | (11.571) | (14.751) |
| lnmar | 0.019 * | | 0.044 *** | | 0.051 ** | |
| | (1.945) | | (3.698) | | (2.133) | |
| lncom | | 0.025 ** | | 0.057 *** | | 0.061 ** |
| | | (2.095) | | (3.089) | | (2.335) |
| Control variables | YES | YES | YES | YES | YES | YES |
| Province FE | | | YES | YES | YES | YES |
| Year FE | | | YES | YES | YES | YES |
| Observations | 270 | 270 | 270 | 270 | 240 | 240 |
| Number of id | | | 30 | 30 | 30 | 30 |
| R-squared | 0.402 | 0.403 | 0.379 | 0.368 | | |
| Sargan | | | | | 2.871 | 18.72 |
| | | | | | (0.720) | (0.711) |
| AR (1) | | | | | −4.990 | −5.414 |
| | | | | | (0.000) | (0.000) |
| AR (2) | | | | | 0.567 | 0.641 |
| | | | | | (0.571) | (0.522) |

Note: Sargan, AR (1), AR (2) are *p*-values in parentheses, mixed regression and fixed regression are t-statistics in parentheses, and the rest are z-statistics; ***, **, * represent the significance levels of 1%, 5%, and 10%, respectively.

Whether using mixed regression or fixed effect regression, it is discovered that command-and-control environmental regulation aids in the promotion of industrial GTFP. However, in the mixed regression, the 5% significance test failed in the market-based environmental regulation to upgrade the development of industrial GTFP, but the mixed regression results are often not robust. From the more robust fixed regression and dynamic panel regression, we found that the promoting effect of market-based environmental regulation on industrial GTFP has passed the 5% significance test. Therefore, it can be concluded that environmental regulation is helpful for the advancement of industrial GTFP. First of all, the cost of non-compliance with environmental regulations is very high, and environmental regulations can contribute to industrial GTFP growth by prompting industrial enterprises to eliminate outdated production capacity. Secondly, environmental regulation can send a clear signal to the outside world: environmental control and production costs in high-pollution industries are being strengthened, which in turn leads to a decrease in investment in high-pollution industries, allowing social capital to enter other green and high-value-added industries. As a result, the industrial structure will be modified, and pollutant emissions will inevitably be decreased, which will ultimately promote the progress of industrial GTFP. In conclusion, environmental control facilitates industrial intensification and green transformation, hence enhancing industrial GTFP.

### 4.2. Robustness Test

Two robustness tests have been carried out to verify the robustness of previous research results in this paper. The first was the model replacement, where the regression model was replaced with a differential GMM. The second was the replacement of key independent variables. There are many ways to express environmental regulation. In this paper, the market-based environmental regulation has been replaced by the ratio of industrial pollution control investment to the total discharge of three industrial wastes, and the command-and-control environmental regulation has been replaced by the ratio of the number of environmental proposals made by the National People's Congress and the Chinese People's Political Consultative Conference to the total amount of industrial waste; the relevant data come from the yearbooks of the provinces. The regression data are shown in Table 3. It shows that whether changing the estimation method or the representation method of the key independent variables, the signs of the coefficients of the market-based

environmental regulation and the command-and-control environmental regulation are all positive, and at least 5% of the coefficients have passed. Therefore, it can be said that both types of environmental regulation can contribute to the progress of industrial GTFP, and the model results are robust.

**Table 3.** Robustness test.

| Variables | Difference Equations (GTFP) | | Replacement Metrics (GTFP) | |
|---|---|---|---|---|
| L.GTFP | 0.755 *** | 0.787 *** | 1.040 *** | 0.867 *** |
| | (8.185) | (7.064) | (12.832) | (25.738) |
| lnmar | 0.121 *** | | 0.020 ** | |
| | (9.991) | | (2.232) | |
| lncom | | 0.101 ** | | 0.349 *** |
| | | (2.187) | | (2.935) |
| Control variables | YES | YES | YES | YES |
| Province FE | YES | YES | YES | YES |
| Year FE | YES | YES | YES | YES |
| Observations | 210 | 210 | 240 | 240 |
| Number of id | 30 | 30 | 30 | 30 |
| Sargan | 6.626 | 1.916 | 2.938 | 1.483 |
| | (0.357) | (0.384) | (0.817) | (0.476) |
| AR (1) | −1.427 | −5.407 | −5.752 | −1.531 |
| | (0.154) | (0.000) | (0.000) | (0.126) |
| AR (2) | 0.326 | 1.199 | 0.437 | 0.160 |
| | (0.745) | (0.230) | (0.662) | (0.873) |

Note: Sargan, AR (1), and AR (2) are *p*-values in parentheses, and z-statistics in other brackets; ***, **, represent the significance levels of 1%, 5%, respectively.

### 4.3. Inspection of External Mechanisms

Models 7 and 9 in Table 4 represent the relationship between innovation capability and heterogeneous environmental regulation. We found that both types of environmental regulation contribute to the advancement of innovation capability. When the government implements regulations, first, the cost of high-emission enterprises will suddenly increase, which will lead to the withdrawal of some high-energy-consuming and low-value-added industries, which will increase the social investment in high-tech industries, and high-tech industries often represent higher technological innovation ability, so environmental regulation improves technological innovation ability. Second, environmental regulation improves the cost of high-emission enterprises. It will not cause all high-emission enterprises to withdraw. Some enterprises with strong financial strength and strong market voice will survive, but enterprises believe that environmental regulation is not a short-term policy. In order to obtain long-term stable profits, enterprises must enhance their innovation capabilities and reduce pollutant emissions. Enterprises are driven by this motivation to improve their technological innovation capabilities. It shows that the "Porter Hypothesis" exists in China, and environmental regulation helps to enhance innovation ability. Models 8 and 10 indicate that environmental regulation and innovation capability are jointly incorporated into the framework of industrial GTFP, and it is found that both environmental regulation and innovation capability can contribute to the progress of industrial GTFP. This proves that environmental regulation contributes to the advancement of technological innovation capabilities, which in turn contributes to the improvement of industrial GTFP. When enterprises try to avoid the rising costs associated with environmental regulation, they will conduct innovation activities, which will eventually lead to the improvement of innovation capability, and the benefits brought by the improvement of innovation capability outweigh the cost of environmental regulation. While increasing the industrial output value, it also reduces the emission of pollutants and carbon dioxide and finally promotes the progress of industrial GTFP, which is consistent with the study by Fan [32].

**Table 4.** External Mechanism Inspection.

| Variables | Market-Based | | Command-and-Control | |
| --- | --- | --- | --- | --- |
| | tech (Model 7) | GTFP (Model 8) | tech (Model 9) | GTFP (Model 10) |
| L.GTFP | | 0.929 *** | | 0.921 *** |
| | | (12.686) | | (23.723) |
| tech | | 0.055 ** | | 0.050 ** |
| | | (2.283) | | (2.084) |
| L.tech | 0.893 *** | | 0.923 *** | |
| | (18.267) | | (28.105) | |
| lnmar | 0.076 *** | 0.048 *** | | |
| | (3.078) | (2.632) | | |
| lncom | | | 0.026 ** | 0.018 *** |
| | | | (2.380) | (2.590) |
| Control variables | YES | YES | YES | YES |
| Province FE | YES | YES | YES | YES |
| Year FE | YES | YES | YES | YES |
| Observations | 240 | 240 | 240 | 240 |
| Number of id | 30 | 30 | 30 | 30 |
| Sargan | 0.448 | 5.124 | 1.068 | 9.236 |
| | (0.799) | (0.925) | (0.785) | (0.416) |
| AR (1) | −5.413 | −4.819 | −1.805 | −1.562 |
| | (0.000) | (0.000) | (0.071) | (0.118) |
| AR (2) | −1.280 | 0.464 | −1.455 | 0.138 |
| | (0.201) | (0.643) | (0.146) | (0.890) |

Note: Sargan, AR (1), AR (2) are *p*-values in parentheses, and z-statistics in other brackets; ***, **, represent the significance levels of 1%, 5%, respectively.

### 4.4. Analysis of Internal Mechanism

Table 5 shows the statistical results of the internal mechanism test. Through models 11, 12, 14 and 15, we found that environmental regulation contributes to increased production efficiency and reduced pollution emissions. Environmental regulations will force enterprises to eliminate outdated production capacity, which is conducive to improving the overall competitiveness of the industrial industry. Finally, the production efficiency of industrial enterprises is improved. At the same time, most environmental administrative penalties and all environmental protection taxes are designed to reduce pollutant emissions. In addition, statistical results also confirm that regulation policies help reduce pollution emissions, which proves the desirability of China's environmental policy. The regression results of Model 13 show that market-based environmental regulation cannot reduce carbon dioxide emissions. The market-based environmental regulation studied in this paper has only included environmental protection tax, and the environmental protection tax does not include the carbon tax. Therefore, China's current environmental protection tax system "reduces pollution without lowering carbon". It shows that market-based environmental regulation based solely on environmental protection taxes will not reduce carbon emissions. Although the carbon trading market has been expanded to the entire nation, it only includes the power industry and not all industrial sectors. Therefore, the carbon trading market must be expanded to include the entire economy. Through Model 16, the result shows that it fails the 5% significance test in that command-and-control environmental regulation can hindrance carbon dioxide emissions, indicating that the primary role of China's command-and-control environmental regulation within the research time range is still to reduce pollution, and the effect of carbon reduction is not very obvious.

**Table 5.** Internal mechanism inspection.

| Variables | Market-Based | | | Command-and-Control | | |
|---|---|---|---|---|---|---|
| | effi (Model 11) | poll (Model 12) | carb (Model 13) | effi (Model 14) | poll (Model 15) | carb (Model 16) |
| L.effi | 0.703 *** | | | 0.744 *** | | |
| | (10.662) | | | (11.912) | | |
| L.poll | | 0.844 *** | | | 0.826 *** | |
| | | (33.444) | | | (33.950) | |
| L.carb | | | 1.288 *** | | | 1.225 *** |
| | | | (16.809) | | | (16.901) |
| lnmar | 0.050 *** | −0.008 *** | 0.042 | | | |
| | (3.077) | (−2.771) | (0.252) | | | |
| lncom | | | | 0.044 ** | −0.010 *** | −0.520 * |
| | | | | (2.258) | (−2.576) | (−1.706) |
| Control variables | YES | YES | YES | YES | YES | YES |
| Province FE | YES | YES | YES | YES | YES | YES |
| Year FE | YES | YES | YES | YES | YES | YES |
| Observations | 240 | 240 | 240 | 240 | 240 | 240 |
| Number of id | 30 | 30 | 30 | 30 | 30 | 30 |
| Sargan | 1.955 | 0.814 | 4.682 | 25.80 | 4.285 | 3.128 |
| | (0.744) | (0.937) | (0.197) | (0.173) | (0.369) | (0.372) |
| AR (1) | −4.522 | −6.215 | −4.082 | −5.027 | −6.208 | −4.446 |
| | (0.000) | (0.000) | (0.000) | (0.000) | (0.000) | (0.000) |
| AR (2) | 0.203 | 0.236 | −0.227 | 0.603 | 0.189 | −0.0257 |
| | (0.840) | (0.813) | (0.821) | (0.547) | (0.850) | (0.979) |

Note: Sargan, AR (1), AR (2) are *p*-values in parentheses, and z-statistics in other brackets; ***, **, * represent the significance levels of 1%, 5%, and 10%, respectively.

### 4.5. Regional Heterogeneity Analysis

For regional heterogeneity studies, researchers in the past frequently divided the country into three parts: east, middle, and west. Although this method is straightforward, it can be kept the same as the national administrative division. However, this method also has drawbacks: the western, middle, and eastern regions represent differences in economic development in China, but this does not mean that every province in the middle or western region belongs to a backward region. It is unreasonable to include the more developed provinces in the middle or western regions into the backward regions for analysis. Therefore, this paper has divided the country into developed regions and backward regions according to the per capita GDP in 2019. The developed regions consist of the top 15 provincial administrative regions, whereas the backward regions consist of the last 15 provincial administrative regions. Table 6 shows the regression results of regional heterogeneity. We found that in economically developed regions, both environmental regulations contribute to the advancement of industrial GTFP. Because the current industrial situation and innovation capabilities of economically developed regions have reached a high level, environmental regulation contributes to the profound adjustment of the industrial structure, eliminates low-end and backward industries, and realizes the green and intensive development level of the industry. However, for the economically backward regions, the market-based environmental regulation significantly impeded the advancement of industrial GTFP, but the command-and-control environmental regulation cannot contribute to the advancement of industrial GTFP because the economically backward regions have not crossed "Porter's turning point", the ability of technological innovation is weak, the ability of industrial agglomeration is poor, and some regions even regard high-energy-consuming industries as the pillar industries of the region. This all results in the poor ability of industrial industries in the region to resist risks; in addition, the government is taxing pollutants or enforcing administrative control. As a result, the profit of the company that is not rich is further compressed, and it can even make the company on the verge of bankruptcy. Thus, it is impossible for the company to achieve transformation and upgrading in a short time, and they are thus ultimately unable to achieve the progress of industrial GTFP.

**Table 6.** Regional heterogeneity analysis.

| Variables | Economic Developed Area | | Economically Backward Areas | |
|---|---|---|---|---|
| | GTFP | GTFP | GTFP | GTFP |
| L.GTFP | 0.818 *** | 0.851 *** | 0.758 *** | 0.752 *** |
| | (8.447) | (9.308) | (7.794) | (7.898) |
| lnmar | 0.052 ** | | −0.025 *** | |
| | (2.131) | | (−3.085) | |
| lncom | | 0.078 ** | | −0.002 |
| | | (2.464) | | (−0.117) |
| Control variables | YES | YES | YES | YES |
| Province FE | YES | YES | YES | YES |
| Year FE | YES | YES | YES | YES |
| Observations | 120 | 120 | 120 | 120 |
| Number of id | 15 | 15 | 15 | 15 |
| Sargan | 10.66 | 26.07 | 16.45 | 16.66 |
| | (0.639) | (0.249) | (0.422) | (0.547) |
| AR (1) | −3.507 | −3.976 | −3.048 | −2.857 |
| | (0.000) | (0.000) | (0.002) | (0.004) |
| AR (2) | 0.777 | 1.063 | 0.974 | 0.702 |
| | (0.437) | (0.288) | (0.330) | (0.483) |

Note: Sargan, AR (1), AR (2) are *p*-values in parentheses, and z-statistics in other brackets; ***, **, represent the significance levels of 1%, 5%, respectively.

## 5. Discussion

Market-based environmental regulation is the government's environmental control by means of taxation, emissions trading and other market instruments [33]. These environmental regulation policies will transfer the cost of the enterprise due to the consumption of the environment from society to the production enterprise, shrink the production boundary of enterprises to a position equal to the optimal production boundary of society, internalize the external problems, and finally realize pollution reduction. It fully reflects the "polluter pays principle". Similarly, command-and-control environmental regulation means that the government directly intervenes in the production and business activities of enterprises through relevant documents, laws and regulations to reduce the pollution discharge of enterprises. Since the cost of enterprises not complying with such environmental regulations is very high, such environmental regulation policies often have the characteristics of a short implementation period and immediate effect. Ultimately, reducing pollution emissions reduces the undesired output in the economic system and realizes the growth of industrial GTFP.

When the "innovation compensation effect" is smaller than the "following cost effect", it means that the cost and revenue loss of the enterprise are greater than the benefit brought by innovation. In the meantime, the enterprise will not choose to innovate, so environmental regulation does not help the advancement of innovation capability [34]. When the "following cost effect" is smaller than the "innovation compensation effect", it means that although the revenue loss and cost of the enterprise increase, the increased benefits through innovation activities are greater. In turn, the increased losses and costs can be compensated by the benefits brought by innovation activities. At this time, enterprises will choose to innovate [35], so environmental regulation can increase innovation. The industrial GTFP represents the intensive development of the industry, and the improvement of innovation means that the industry develops in the direction of intensification, thereby improving the industrial GTFP. For China, the situation faced by enterprises is not only the two situations mentioned above. When the cost of a company in response to environmental regulations increases, the innovation effect will be greater than the cost-effectiveness on the leading enterprises due to their substantial capital and technical strength, and they will improve themselves through R&D investment. However, for enterprises with relatively backward technology, environmental regulation will increase their costs. Suppose they

do not have strong financial and technical strength. In that case, their cost effect will be greater than the innovation effect, and they will inevitably withdraw from the market in the face of competition from large enterprises and environmental regulation by the government. At the same time, environmental regulation will cause social capital to reduce investment in high-energy-consuming and high-emission industrial projects and instead invest in high-tech enterprises, which will also promote the improvement of innovation capabilities. To sum up, environmental regulation allows enterprises without innovation ability to withdraw from the market, while it allows enterprises with innovation ability to innovate and attract social capital to enter high-tech enterprises, thereby enhancing technological innovation ability and industrial GTFP.

The industrial GTFP describes the development of the industry systematically. In terms of output, it not only reflects industrial economic development but also encompasses emissions of pollutants and carbon dioxide. The high-quality development of industry also involves all aspects of the economy and the environment. In this paper, industrial GTFP has been decomposed into three parts: industrial production efficiency, pollution emission, and carbon dioxide emission according to its source. In addition, the influence of heterogeneous environmental regulation on each decomposed part has been analyzed, and then, the internal transmission mechanism of environmental regulation affecting industrial GTFP has been discussed. Environmental regulation will compel enterprises to eliminate backward production methods, transform and upgrade, and transition from extensive to intensive development mode, all of which will contribute to an overall increase in production efficiency [36]. At the same time, environmental regulation is a government management method aimed at environmental pollution, and the cost of companies not complying with environmental regulations is very high. Therefore, when companies face the government's environmental regulations, they will inevitably reduce the discharge of pollutants. Since the current market-based environmental regulation policy focuses primarily on taxing pollutants and does not include a carbon tax, market-based environmental regulations cannot be used to reduce carbon emissions.

## 6. Conclusions and Policy Recommendations

### 6.1. Conclusions

This paper used the industrial panel data of various provinces in China from 2011 to 2019. On the basis of calculating the industrial GTFP through the SBM-GML index, the system GMM estimation method has been used to empirically analyze the effect of market-based environmental regulation and command-and-control environmental regulation on industrial GTFP. In addition, the internal and external mechanism routes by which heterogeneous environmental regulation influences industrial GTFP have been investigated. Statistics show that all environmental regulations are helpful for the progress of industrial GTFP and pass the robustness test. The external mechanism test found that environmental regulation contributes to the advancement of technological innovation capabilities, which in turn contributes to the improvement of industrial GTFP. In addition, this paper has decomposed industrial GTFP into three parts: production efficiency, pollution emissions and carbon emissions. Then, we have studied the internal transmission effect of heterogeneous environmental regulation on industrial GTFP and found that both market-based environmental regulation and command-and-control environmental regulation can promote the progress of industrial production efficiency and significantly inhibit industrial pollution emissions, but the market-based environmental regulation does not have an effective impact on carbon emissions. Finally, this paper divided the country into developed regions and backward regions based on per capita GDP in 2019, and it conducted regional heterogeneity analysis. The statistical results found that environmental regulation in economically developed regions can promote the growth of industrial GTFP, but for economically backward regions, the market-based environmental regulation significantly inhibits the progress of industrial GTFP, while the command-and-control environmental regulation cannot contribute to the advancement of industrial GTFP.

*6.2. Policy Recommendations*

According to the research content and results, this paper has proposed the following policy suggestions: (1) Establish and improve the market-based environmental regulation policy system. Through the analysis of the decomposition part of industrial GTFP, as can be seen, the current market-based environmental regulation has the effect of "reducing pollution without reducing carbon", and the market-based environmental regulation mainly based on environmental protection tax does not restrict carbon dioxide emissions. China's current market-based environmental regulations urgently need to be updated in terms of carbon tax and carbon trading market. Although China's carbon trading market has begun to operate, the included trading groups only include power companies and have not expanded to the entire industry. Therefore, it is necessary to incorporate the entire industry into the carbon trading market system as soon as possible and pay attention to the use of market-based regulation. (2) Improve the ability of technological innovation. We found that in addition to acting as a mechanism variable for environmental regulation to affect industrial GTFP, the technological innovation capability itself also promotes the improvement of industrial GTFP. So, we should increase investment in technology, lead the industrial industry to develop in the direction of high added value and high technology, better adjust the industrial structure, and reduce dependence on high energy-consuming industries. (3) Economically backward areas should adhere to the policy of paying equal attention to economic development and pollution control. Environmental regulation policies in economically backward areas will not promote the progress of industrial GTFP, because the industrial structure is relatively simple, and the industrial chain has a poor ability to resist risks. This area should increase investment promotion, actively adjust the industrial structure, improve the industrial chain in the region, remove the industrial structure dominated by high energy consumption, and strengthen environmental governance.

**Author Contributions:** Conceptualization, G.Y., L.J. and C.X.; Methodology, L.J.; Data curation, L.J.; Software, L.J.; Formal analysis, L.J.; Resources, G.Y. and C.X.; Project administration, G.Y. and C.X.; Supervision, G.Y. and C.X.; Writing—original draft, L.J.; Writing—review and editing, G.Y. and C.X.; All authors have read and agreed to the published version of the manuscript.

**Funding:** This paper is supported by the Key R&D Program of Shandong Province, China (Grant No.: 2021RZA01018), Natural Science Foundation Project of Shandong Province, China (Grant No.: ZR2019PD014) and the Major Innovation Projects of Science, Education, and Industry Integration of Qilu University of Technology (Shandong Academy of Science), (Grant No.: 2022JBZ02-05).

**Institutional Review Board Statement:** Not applicable.

**Informed Consent Statement:** Not applicable.

**Data Availability Statement:** The data involved in this study are all from public data.

**Conflicts of Interest:** The authors declare no conflict of interest.

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
