# Peer review of "How Environmental Regulation Affects Industrial Green Total Factor Productivity in China: The Role of Internal and External Channels"

_sustainability, doi:10.3390/su142013500_

Round 1

Reviewer 1 Report

I suggest that authors take note of the suggestions below:

Abstract:
Authors should briefly outline the: (1) research problem, (2) methodology (theoretical and empirical part, research methods and techniques), (3) added value of the research, and (4) added value of the article from the professional/scientific/political point of view.

Chapter: 1. Introduction
Authors should start by clearly stating the research problem and explaining why the research problem is worth exploring. In this chapter, it should be made clear what the added value of the research and what is the added value of the article is.

Reviewer 2 Report

The article is very good. In some of the next publications the authors should highlight the role of the educational system to prepare future managers who to implement the given proposals.

Author Response

We are very grateful to expert for reviewing the manuscript so carefully. We have read your comments carefully and we feel that your comments are very important for our future academic research.

Point 1: The article is very good. In some of the next publications the authors should highlight the role of the educational system to prepare future managers who to implement the given proposals.

Response 1: We will seriously consider your suggestions on the education system as a research direction.

Thank you again for your review!

Reviewer 3 Report

The presented article touches on an interesting and important topic and contains relevant scientific results. At the same time, there are a number of limitations that complicate the correct understanding of the article and the authors' contribution to the solution of the stated research objective. The following recommendations can be suggested to improve the article. It is recommended to shorten the title of the article. It is recommended to edit the abstract in order to more clearly and fully reflect all the results of the study. There is excessive use of abbreviations in the abstract and keywords. It is recommended to separate the introduction and the literature analysis (theoretical background). This will help to present each of these sections more clearly and logically. The methodology of the study is not fully disclosed (both in terms of methods and in terms of the justification of the indicators selected for evaluation and calculations). More attention should be paid to the author's contribution to the formation of the methodology and the results of the study. The article uses abbreviations without prior full mention of the term (or model) and its characteristics. In addition, references to literature sources are given in the text in the format of numbers, but the list of literature sources at the end of the article does not contain numbers. The authors' statements on lines 158-159 require clarification (are the statements considered in pairs?). In the study, the authors indicate that data is used from 2010 to 2019. At the same time, the article states that "a nationwide carbon trading market was not established until 2021". In this regard, the results of the relevant part of the analysis raise questions. It is required to improve the English language in the article.

Round 2

Reviewer 3 Report

Thank you for the detailed feedback and consideration of recommendations.